# Integrated Bioinformatics and Validation Reveal *IFI27* and Its Related Molecules as Potential Identifying Genes in Liver Cirrhosis

**DOI:** 10.3390/biom14010013

**Published:** 2023-12-21

**Authors:** Zhiyu Xiong, Ping Chen, Mengqin Yuan, Lichao Yao, Zheng Wang, Pingji Liu, Yingan Jiang

**Affiliations:** Department of Infectious Diseases, Renmin Hospital of Wuhan University, Wuhan 430060, China; 2022203020019@whu.edu.cn (Z.X.); ching1101@whu.edu.cn (P.C.); 2018283020183@whu.edu.cn (M.Y.); yaolichao09@whu.edu.cn (L.Y.); gywangzheng@whu.edu.cn (Z.W.); 2021283020133@whu.edu.cn (P.L.)

**Keywords:** liver cirrhosis, mitochondrial dysfunction, macrophages, oxidative stress, bioinformatics analysis, machine learning

## Abstract

Liver cirrhosis remains a significant global public health concern, with liver transplantation standing as the foremost effective treatment currently available. Therefore, investigating the pathogenesis of liver cirrhosis and developing novel therapies is imperative. Mitochondrial dysfunction stands out as a pivotal factor in its development. This study aimed to elucidate the relationship between mitochondria dysfunction and liver cirrhosis using bioinformatic methods to unveil its pathogenesis. Initially, we identified 460 co-expressed differential genes (co-DEGs) from the GSE14323 and GSE25097 datasets, alongside their combined datasets. Functional analysis revealed that these co-DEGs were associated with inflammatory cytokines and cirrhosis-related signaling pathways. Utilizing weighted gene co-expression network analysis (WCGNA), we screened module genes, intersecting them with co-DEGs and oxidative stress-related mitochondrial genes. Two algorithms (least absolute shrinkage and selection operator (LASSO) regression and SVE-RFE) were then employed to further analyze the intersecting genes. Finally, *COX7A1* and *IFI27* emerged as identifying genes for liver cirrhosis, validated through a receiver operating characteristic (ROC) curve analysis and related experiments. Additionally, immune infiltration highlighted a strong correlation between macrophages and cirrhosis, with the identifying genes (*COX7A1* and *IFI27*) being significantly associated with macrophages. In conclusion, our findings underscore the critical role of oxidative stress-related mitochondrial genes (*COX7A1* and *IFI27*) in liver cirrhosis development, highlighting their association with macrophage infiltration. This study provides novel insights into understanding the pathogenesis of liver cirrhosis.

## 1. Introduction

Liver cirrhosis, stemming from chronic liver injury induced by various hepatotoxic factors, poses a severe global health concern. It may progress into liver failure and even hepatocarcinoma, claiming approximately two million lives annually worldwide [1,2]. Despite liver transplantation being the most effective remedy for end-stage liver diseases, its efficacy is constrained by donor shortage, immune rejection risks, surgical complexities and high cost. Therefore, an in-depth investigation of the precise cellular and molecular mechanisms underlying liver cirrhosis holds immense promise for mitigating its progression and fostering the development of more potent therapeutic strategies.

Oxidative stress refers to the disruption in the equilibrium between oxidants and antioxidants, leading to an accumulation of excessive oxidative substances within cells, ultimately causing cell and tissue damage [3]. Recent studies underscore oxidative stress and mitochondrial dysfunction as potent contributors to liver cell death and tissue damage. Valgimigli et al. observed increased ROS levels in patients with chronic hepatitis C compared to healthy individuals, with a positive correlation observed between ROS levels and histological disease activity [4]. Additionally, studies elucidate that dysfunctional mitochondria play a crucial role in non-alcoholic fatty liver disease (NAFLD) progression towards cirrhosis by promoting liver lipid accumulation and triggering inflammation [5]. For example, Koliaki et al. identified increased mitochondrial ROS production in patients with NASH compared to those with pure steatosis, where an abundance of ROS activates inflammatory signaling pathways such as NF-κB and JNK, inducing the expression of inflammatory cytokines like TGF-β and TNF-α, thereby accelerating NAFLD progression [6]. Collectively, various mitochondria-related factors, including ROS overproduction, oxidative stress-mediated cellular damage and inflammatory factor-related cascades, contribute significantly to liver cirrhosis development, necessitating further exploration into the relationship between cirrhosis and mitochondria dysfunction.

Recent advancements in high-throughput sequencing technology have provided a robust avenue for studying liver disease pathogenesis and characteristics [7]. In this study, we screened liver cirrhosis-related gene matrices and ROS-related mitochondrial gene information from the Gene Expression Omnibus (GEO) database, MitoCarta3.0 database and GeneCard database. Subsequently, we intersected co-expressed differential genes (co-DEGs) from three different liver cirrhosis datasets, module genes identified via the weighted gene co-expression network analysis (WGCNA) algorithm and oxidative stress-related mitochondrial genes. The overlapping genes were analyzed using two machine learning algorithms, leading to the identification of potential liver cirrhosis identifying genes. Furthermore, employing the CIBERSORT algorithm, we scrutinized 22 immune cell types associated with liver cirrhosis, correlating them with identifying genes to unravel the underlying molecular mechanisms fueling liver cirrhosis development. Our investigation revealed increased levels of both *IFI27* and *COX7A1* in liver cirrhosis. Particularly, *IFI27* displayed a higher expression than *COX7A1* in M1 macrophages, exhibiting a positive correlation with M1 macrophage polarization. These findings suggest *IFI27*’s potential involvement in M1 macrophage polarization during liver cirrhosis, highlighting its significance in liver cirrhosis progression.

## 2. Materials and Methods

### 2.1. Data Collection

Microarray datasets (GSE14323 and GSE25097) were sourced from the GEO database (http://www.ncbi.nlm.nih.gov/geo/, accessed on 28 October 2023) [8]. GSE14323 comprised 41 cirrhotic liver samples and 19 controls, while GSE25097 included 40 cirrhotic and 6 healthy human liver samples. We merged the two datasets to form a comprehensive new dataset by using the R packages “Limma” and “sva”. Additionally, 884 oxidative stress-related mitochondrial genes (Appendix A in Appendix A) were gathered from the MitoCarta3.0 database (http://www.broadinstitute.org/mitocarta, accessed on 28 October 2023) and the GeneCard database (http://www.genecards.org/, accessed on 28 October 2023) [9,10].

### 2.2. Differential Expression Analysis

Differentially expressed genes (DEGs) between the liver cirrhosis and healthy groups in the three datasets were identified using the R package ‘Limma’. Criteria for differential expression were an absolute value of log2 (fold change) (log2FC) > 0.585 and FDR < 0.05. Visualization of downregulated and upregulated DEGs was accomplished using the R package ‘ggplot2′ for volcano plots [11] and ‘pheatmap’ for heatmap representation. Co-DEGs across the three datasets were determined with the ‘VennDiagram’ R package [12].

### 2.3. Functional Enrichment Analysis

To analyze the biological functions and pathways associated with co-DEGs, Gene Ontology (GO) and Kyoto Encyclopedia of Genes and Genomes (KEGG) analyses were performed using the R package ‘clusterProfiler’. Significant thresholds for enrichment analysis were set at q < 0.05. To further uncover the biological differences between the liver cirrhosis and control samples, we subjected the co-DEGs to gene set enrichment analysis (GSEA) using the R package ‘clusterprofiler’ [13], with a significance level set at adjusted q < 0.05. Moreover, Disease Ontology (DO) combined with GO was used to elucidate disease–gene interactions using the Human Disease Ontology (www.disease-ontology.org, accessed on 28 October 2023) database [14].

### 2.4. WGCNA

WGCNA analyses co-express gene modules that have high biological significance and explore the relationship between gene networks and diseases [15]. Therefore, the R package ‘WGCNA’ was used to identify liver cirrhosis-related module genes across the three databases. Shared genes among co-DEGs, module genes and ROS-related mitochondrial genes were determined using the R package ‘VennDiagram’.

### 2.5. Identification and Validation of Candidate Identifying Genes

The LASSO logistic regression was employed to further screen the candidate identifying genes associated with liver cirrhosis and oxidative stress. The R package ‘glmnet’ was used to perform LASSO logistic regression and minimal lambda was considered optimal [16]. Additionally, the SVM-RFE algorithm conducted using the R packages ‘e1071’ and ‘caret’ calculated the point with the smallest cross-validation error, thereby screening candidate identifying genes [17]. Subsequently, overlapping genes were further validated using GSE89377 from the GEO database. ROC curves and area under the curve (AUC) calculations made using the R package ‘pROC’ were used to assess gene sensitivity and specificity [18].

### 2.6. Analysis of Immune Cell Infiltration

The infiltration levels of 22 immune cell types based on the merged dataset were estimated using the CIBERSORT algorithm. CIRBERSORT is a deconvolution algorithm used to characterize the composition of the immune cells in a tissue [19]. The Wilcoxon test was used to evaluate statistical significance (*p* < 0.05) and ‘heatmap’ and ‘vioplot’ in R were used to visualize immune cell infiltration differences between liver cirrhosis and healthy control liver samples. Furthermore, principal component analysis (PCA) using the R package ‘scatterplot3d’ was employed to visualize high-dimensional gene expression data [20]. Moreover, Spearman correlation analysis was used to verify the association between the candidate identifying genes and immune infiltration [21]. The results were visualized using the R package ‘corrplot’.

### 2.7. Mice Model Preparation

Male C57BL/6 mice (8 weeks, 17–22 g) were randomly assigned to the liver cirrhosis group and normal control group. All animal experiments were performed per the institutional guidelines and approved by the Animal Care and Use Committee of Wuhan University. Liver cirrhosis was induced in mice using TAA dissolved in sterile physiological saline via intraperitoneal injection (200 mg/kg) twice per week for a duration of three months, while the control group received sterile physiological saline.

### 2.8. Histopathology

Liver tissue samples from mice were fixed with a 4% paraformaldehyde solution, embedded in paraffin and meticulously sectioned into slices (5 µm thick). To elucidate general histopathological characteristics, H&E staining, Masson staining and Sirius red staining were used to evaluate the histopathological variations in the liver.

### 2.9. Tissue Immunofluorescent Staining

To verify *IFI27* expression in liver macrophages, the mice’s liver sections were fixed in formalin, permeabilized in 0.1% Triton X-100 and blocked with 10% bovine serum albumin. The samples were subsequently incubated overnight with specific primary antibodies (F4/80, Servicebio GB113373, 1:1000), (*IFI27*, Affinity, DF8989, 1:100) at 4 °C. Following this, secondary antibodies were incubated at 25 °C for 1 h. Nuclei were stained with DAPI for 10 min under dark conditions and visualized. The images were analyzed and quantified via ImageJ software (Version 1.5.4).

### 2.10. Cell Culture and Experimental Design

The mice macrophage cell line RAW264.7 was obtained from the cell bank of the Chinese Academy of Sciences (Shanghai, China). RAW264.7 macrophages were cultured in DMEM (HyClone, Logan City, UT, USA) supplemented with 10% fetal bovine serum (Gibco, California State City in California, USA) and 1% penicillin–streptomycin in a humidified 5% CO2 incubator at 37 °C. The RAW264.7 macrophages were subsequently stimulated with LPS (1 μg/mL) for 24 h to induce the differentiation of the M1 phenotype. The untreated RAW264.7 cells were recognized as the M0 phenotype.

### 2.11. Small Interfering RNA (siRNA) Transfection

Specific siRNA targeting *IFI27* (si-*IFI27*) or negative control siRNA (si-NC) transfection was performed using Lipo 3000 (Invitrogen, Carlsbad City, CA, USA) according to the manufacturer’s protocols. After 24 h of transfection, we added LPS (1 μg/mL) into the media for the si-*IFI27* group and si-NC group separately for 24 h. The *IFI27* target sequence was GCACUGAAGGUUGGCACCAUUTT (si-1).

### 2.12. Total RNA Extraction and Real-Time Quantitative Polymerase Chain Reaction (RT-qPCR)

The total RNA was extracted from mice liver tissues and RAW264.7 cells using Trizol reagent (Invitrogen, USA). RNA was converted to cDNA using a cDNA synthesis kit (Servicebio, Wuhan, China). RT-qPCR was performed using Power Up SYBR Green Master Mix on a real-time system. Each reaction mixture (20 μL) comprised 8 μL of cDNA, ddH2O water, 1 μL each of forward and reverse primers and 12 μL SYBR Green Master Mix. The applied cycle conditions were 50 °C for 2 min and 95 °C for 2 min followed by 40 cycles of 15 s at 95 °C and 1 min at 60 °C. Herein, GAPDH was applied in the qPCR normalization. The results were preliminarily analyzed using the LightCycler^®^96 software. Furthermore, the 2-ΔΔCt method was used to calculate the relative mRNA level of the target gene in each sample. GraphPadPrism8.0 was used to identify statistical differences between groups. All primers used are listed below (Table 1).

### 2.13. ELISA and ROS Measurement

RAW 264.7 cells were seeded into 96-well plates at a density of 1 × 10^6^ cells/mL. The M1 phenotype and transfection cell model were stimulated with LPS as described above. After stimulation with LPS for 24 h, the supernatants were collected for measuring the protein levels of IL-1β and TNF-α using IL-1β and TNF-α kits (Invitrogen, Carlsbad City, CA, USA), respectively. Cellular ROS content was detected in RAW264.7 cells using a Reactive Oxygen Species Assay kit (Beyotime, Wuhan, China), following the manufacturer’s instructions.

### 2.14. Statistical Analysis

All data were analyzed with GraphPad Prism 6.0 and presented as mean ± standard error of the mean. Student’s *t*-tests or one-way ANOVAs were used to compare groups and the Spearman method was used to assess correlations. *p* < 0.05 was considered significant.

## 3. Results

### 3.1. Identification of DEGs

DEGs between liver cirrhosis and healthy liver samples were identified at a threshold of |logFC| > 0.585, FDR < 0.05, revealing 1789, 1190 and 944 DEGs in the GSE14323, GSE25097 and merged datasets, respectively (Figure 1A–C). Figure 1D presents the 460 overlapping co-DEGs among the three databases. Of these, 325 genes were upregulated while 135 genes were downregulated.

### 3.2. Functional Enrichment Analysis

To explore the biological functions and related disease targets of co-DEGs, GO, KEGG, GSEA and DO analyses were conducted. Firstly, we annotated the GO and KEGG functions of co-DEGs. Regarding biological processes (BP) and cellular components (CC), co-DEGs were primarily enriched in cytokine-mediated signaling, leukocyte chemotaxis and collagen containing. Molecular functions (MF) revealed enrichment in mainly the ECM (Figure 2A,B). These co-DEGs were also enriched in KEGG pathways related to cytokine signaling, cell–cell adhesion regulation and cell chemotaxis (Figure 2C,D).

Furthermore, DO enrichment analysis illustrated that the co-DEGs were mainly related to hepatitis, pulmonary disease and arteriosclerosis (Figure 3A,B). Moreover, GSEA shows the following BP between groups: KEGG pathways including N-glycan biosynthesis, peroxisome, protein export, retinol metabolism and valine, leucine, and isoleucine degradation were enriched in control samples (Figure 3C), whereas cell adhesion molecules, chemokine signaling, cytokine receptor interaction, ECM receptor and focal adhesion were significantly enriched in the liver cirrhosis group (Figure 3D). These findings highlight the crucial role of co-DEGs in cytokine signaling and fibrotic-related diseases.

### 3.3. WGCNA

WGCNA of the top 1083 varied genes identified two modules at a soft threshold power of 10 (Figure 4A,B). The turquoise module, comprising 890 genes, showed a strong correlation with liver cirrhosis, while the blue module, comprising 193 genes, exhibited a lower correlation. Figure 4C presents the strong relationship between module membership in the turquoise module and gene significance for liver cirrhosis (cor = 0.89). Therefore, the turquoise module served as the key module in the subsequent analysis.

Subsequently, we obtained 884 mitochondrial genes related to oxidative stress from the MitoCara3.0 database and GeneCard database. Venn maps identified 21 overlapping genes among co-DEGs, key module genes and oxidative stress-related mitochondrial genes (Figure 4D).

### 3.4. Exploring Candidate Identifying Genes Using LASSO Regression and SVM-RFE

The LASSO logistic regression algorithm and SVM-RFE algorithm, based on the 21 genes mentioned above (Figure 5A,B), revealed eight identifying genes (*PC*, *IDH2*, *GLUD1, EFHD1*, *ACACB*, *COX7A1*, *MTHFD2* and *IFI27*) via LASSO and two genes (*COX7A1* and *IFI27*) via SVM-RFE. Finally, two oxidative stress-related mitochondrial genes *COX7A1* and *IFI27* were selected as identifying genes associated with liver cirrhosis (Figure 5C).

### 3.5. Validation of Candidate Identifying Genes

Compared with the control samples, the expression levels of *COX7A1* and *IFI27* were significantly upregulated in the liver cirrhosis samples in both the merged dataset and validation dataset GSE89377 (Figure 5D–G). As shown in the ROC curve analysis, the AUC values of *COX7A1* and *IFI27* were 0.983, 0.998, 0.840 and 0.763 in the merged dataset and GSE89377, respectively (Figure 5H–K). These results demonstrate the high accuracy of *IFI27* and *COX7A1* as identifying genes.

### 3.6. IFI27 and COX7A1 Are Upregulated in Liver Cirrhosis

To further elucidate the potential implications of *IFI27* and *COX7A1* in liver cirrhosis, we established a TAA-induced mouse liver cirrhosis model. A histopathological assessment using H&E staining, Masson staining and Sirius red staining was used to measure the degree of liver cirrhosis (LF). Compared to the healthy control group, the fibrous area of the liver tissue was significantly increased in the LF group treated with TAA (Figure 8A–C). Subsequently, RT-PCR analysis of the liver tissue revealed increased *IFI27* and *COX7A1* expression in the TAA-treated cirrhosis group compared to the control group (Figure 8E).

### 3.7. Analysis of Immune Cell Infiltration

To comprehensively understand the correlations between liver cirrhosis genes and immune cells, 22 immune cells were further studied using the CIBRSORT algorithm (Figure 6A). Significant correlations were observed between liver cirrhosis expression and T cells as well as macrophages. Figure 6B illustrates the correlation value between infiltrated immune cells and indicates the correlation degree by score. Through the intersection of the difference and correlation analyses, M0 and M1 macrophages were positively correlated with the liver cirrhosis samples (Figure 6C). PCA further indicated potential distinctions in immune infiltration between the liver cirrhosis and healthy control liver groups (Figure 6D).

Furthermore, correlation analysis confirmed positive associations of the two identifying genes (*COX7A1* and *IFI27*) with 22 immune cells (Figure 7A,B). *COX7A1* showed positive correlations with M0 macrophages (R = 0.38, *p* = 5.1 × 10^−0.5^) and M1 macrophages (R = 0.433, *p* = 5.2 × 10^−0.6^) (Figure 7C,D). Simultaneously, *IFI27* was also positively correlated with M0 (R = 0.39, *p* = 3.8 × 10^−0.5^) and M1 macrophages (R = 0.61, *p* = 3.5 × 10^−12^) (Figure 7E,F). Collectively, these data suggest that aberrant *COX7A1* and *IFI27* expression impacts immune activity related to liver cirrhosis, identifying them as promising identifying genes for understanding this association.

### 3.8. IFI27 Is Crucial for M1 Macrophage Polarization and ROS Production

M1 macrophages, particularly pro-inflammatory M1-like macrophages, play a pivotal role in liver damage and cirrhosis pathogenesis [22,23,24,25]. RT-PCR analysis revealed significantly increased expression of *IFI27* in M1-like macrophages compared to low expression of *COX7A1* in RAW264.7 (Figure 8F). To validate this association, tissue immunofluorescent staining was performed, which revealed the co-localization of *IFI27* with macrophages (Figure 8D). Notably, increased *IFI27* expression was observed in both the LF group and M1-like macrophages.

To further substantiate the hypothesis that *IFI27* could promote macrophage polarization towards the M1-like phenotype, we used specific siRNA targeting *IFI27* to knock down its expression in the RAW264.7 cells (Figure 8G). A significant reduction in major M1 polarization identifying genes, including CD86, CD80 and iNOS, was observed in the si-*IFI27* group (Figure 8I–K). Additionally, we tested two main inflammation factors (IL-1β and TNF-α) in the different groups to validate the above results. The PCR (Figure 8L,M) and ELISA (Figure 8H) results demonstrated that the levels of IL-1β and TNF-α were significantly decreased in the si-*IFI27* group compared to the LPS group. Moreover, comparing ROS expression among the different groups revealed that the siRNA *IFI27* treatment group significantly downregulated the ROS expression levels of M1 macrophages (Figure 8N,O). Collectively, these findings further support *IFI27*’s pivotal role in macrophage polarization, suggesting its potential to promote transitions toward the M1-like subtype while concurrently impacting ROS production.

## 4. Discussion

Liver cirrhosis, a precursor to cirrhosis, manifests as a dysregulated balance in ECM synthesis and degradation, leading to its abnormal accumulation in the liver. Activated HSCs can transdifferentiate into myofibroblast-like cells, generating excess ECM, inflammatory cytokines and chemokines. This triggers the recruitment and activation of quiescent HSCs, enhancing inflammatory responses and driving cirrhosis progression [26]. Despite liver transplantation’s therapeutic limitations, there is an urgent need to comprehend the pathogenesis of hepatic cirrhosis and explore innovative interventions.

Liver cirrhosis develops due to multifaceted factors including the accumulation of multiple factors such as cascading inflammatory responses, immune system imbalance and oxidative stress [27]. Notably, the mitochondria’s pivotal role in oxidative stress is tied to its dysfunction in inflammasome activation, apoptosis, immune response and cirrhosis [28]. Chronic liver injury induces excessive mitochondrial ROS production, interfering with the mitochondrial respiratory chain (MRC), leading to mitochondrial dysfunction and programmed cell death pathway activation in liver cells [29]. Furthermore, MRC damage can also disrupt calcium homeostasis, leading to increased mitochondrial membrane permeability and liver cell death [30]. Recent research links ferroptosis, mitochondrial dysfunction and ROS accumulation, emphasizing their roles in liver cirrhosis progression [31,32,33]. Therefore, exploring the mitochondria–liver cirrhosis relationship offers a new perspective on understanding its mechanisms.

This study analyzed mRNA microarray datasets (GSE14323, GSE25097 and combined) to identify 460 co-DEGs, with 325 downregulated and 135 upregulated genes. Functional enrichment analysis revealed that these co-DEGs were mainly enriched in collagen formation, cytokine-mediated signaling pathways and leukocyte chemotaxis, all pivotal processes in liver cirrhosis development. For example, TGF-β1 signaling promotes the transdifferentiation of HSCs into myofibroblast-like cells, leading to the production of large amounts of collagen and overexpression of TIMPs, consequently leading to the accumulation of ECM [34]. Moreover, the TGF-β1 signaling pathway also participates in hepatic cell death and lipid accumulation [35]. Leukocytes also play a role in promoting liver cirrhosis by secreting myeloperoxidase (MPO), which can activate HSCs [36]. Therefore, the chemotaxis of leukocytes is speculated to be involved in the pro-fibrotic effects of liver cirrhosis. To further identify module genes highly related to liver cirrhosis, we analyzed the merged dataset using WGCNA. In addition, we used two algorithms (LASSO and SVM-RFE) to further filter the intersecting genes. The overlapping genes were intersected from module genes, co-DEGs and ROS-related mitochondrial genes obtained from the MitoCarta3.0 database and GeneCard database. Finally, *COX7A1* and *IFI27* were identified as potential candidate identifying genes.

*COX7A1*, a cytochrome c-oxidase subunit, plays an important role in the super-assembled complex of the mitochondrial electron transport chain. Its overexpression enhances the activity of tricarboxylic acid and complex IV in the mitochondrial ETC, inducing calcium homeostasis dysregulation and promoting hepatocyte apoptosis [37]. It also accumulates autophagosomes by blocking mitophagy flux, downregulating PGC-1 and upregulating NOX2, which results in the activation of HSCs and promotion of liver cirrhosis development [38]. *IFI27*, a small hydrophobic protein enriched in mitochondria, participates in diverse BP [39]. *IFI27* can inhibit SLC7A11 by regulating the cell cycle protein P53, leading to the inhibition of System XC- and consequently promoting ferroptosis [40]. Moreover, *IFI27* also causes mitochondrial membrane destabilization and serves as a tandem gene for ferroptosis and apoptosis. Additionally, it activates the mitochondrial stress signaling pathway, leading to Bcl-2-associated protein X (BAX)/Bcl-2 homologous Antagonist Killer (BAK) oligomerization. The BAK/BAK complex causes mitochondrial dysfunction by disrupting the mitochondrial membrane potential, releasing a large amount of cytochrome c from the mitochondria and forming apoptosomes. Consequently, this leads to the activation of caspase-2, 3, 9, which further enhances hepatocyte apoptosis [41]. In this study, *IFI27* and *COX7A1* were observed to exhibit higher expression levels within liver cirrhosis samples compared to the normal group. Therefore, these mitochondrial oxidative stress-related genes, *COX7A1* and *IFI27*, have pivotal roles in the development of liver cirrhosis.

Understanding immune cell infiltration’s impact on liver cirrhosis, we utilized the CIRBERSORT algorithm to analyze 22 immune cell proportions, revealing a strong association between macrophages and cirrhosis-related genes. Hepatic macrophages, comprising Kupffer cells and monocyte macrophages derived from bone marrow, are crucial immune cells in the development of liver inflammation and cirrhosis [42]. In liver injury, these macrophages transform from M0 to M1 pro-inflammatory cells due to DAMP release and ROS generation [42]. The interplay between hepatic macrophages and HSCs forms a positive feedback loop fueling liver cirrhosis. For example, activated hepatic macrophages produce cytokines, such as IL-6, IL-1β, TNF-α and macrophage colony-stimulating factor (M-CSF), which sustain their pro-cirrhosis and pro-inflammatory activities while also stimulating HSCs to form an inflammatory cascade [23]. Cai et al. demonstrated through in vivo experiments that CXCL6 secreted by Kupffer cells induces the expression of pro-fibrotic genes [43]. ROS can trigger an inflammatory cascade and aggravate liver cirrhosis and HSC-driven ECM production by accelerating the activation of hepatic macrophages. Tantawy et al. showed that *IFI27* can play a pro-inflammatory role by inducing the infiltration of lung macrophages [44]. However, the relationship between *IFI27* and *COX7A1* and hepatic macrophages remains unexplored.

Our experimental findings indicate elevated *IFI27* expression during M1 polarization, while *COX7A1*’s association with macrophages might be less prominent. Subsequently, we conducted a series of follow-up validation experiments with a primary emphasis on *IFI27*. These experiments conclusively confirmed that *IFI27* induces macrophages to transition towards an M1-like phenotype while simultaneously promoting the production of pro-inflammatory factors and oxidative stress. Thus, we speculate that *IFI27* likely plays a central role in modulating hepatic macrophage polarization, influencing liver cirrhosis progression. Further investigation is warranted to fully elucidate *IFI27*′s impact on M1 macrophage polarization.

Furthermore, several limitations in the study must be acknowledged. While our data source relies on public databases, limited sample sizes could influence result precision and reliability. Moreover, the analyses were based on bioinformatics algorithms without original sequencing data, necessitating further verification before clinical application. Finally, comprehending the precise molecular mechanisms of mitochondrial oxidative stress-related candidate genes in liver cirrhosis pathogenesis demands additional in vivo and in vitro research.

## 5. Conclusions

This study combined bioinformatics analysis and machine learning to identify mitochondrial oxidative stress-related genes in the development of liver cirrhosis. We found that *COX7A1* and *IFI27* may be potential candidate identifying genes, and immune cell infiltration analysis elucidated the strong correlation of macrophages in the development of liver cirrhosis. These findings provide a new perspective for understanding the pathogenesis of liver cirrhosis.

## Figures and Tables

**Figure 1 biomolecules-14-00013-f001:**
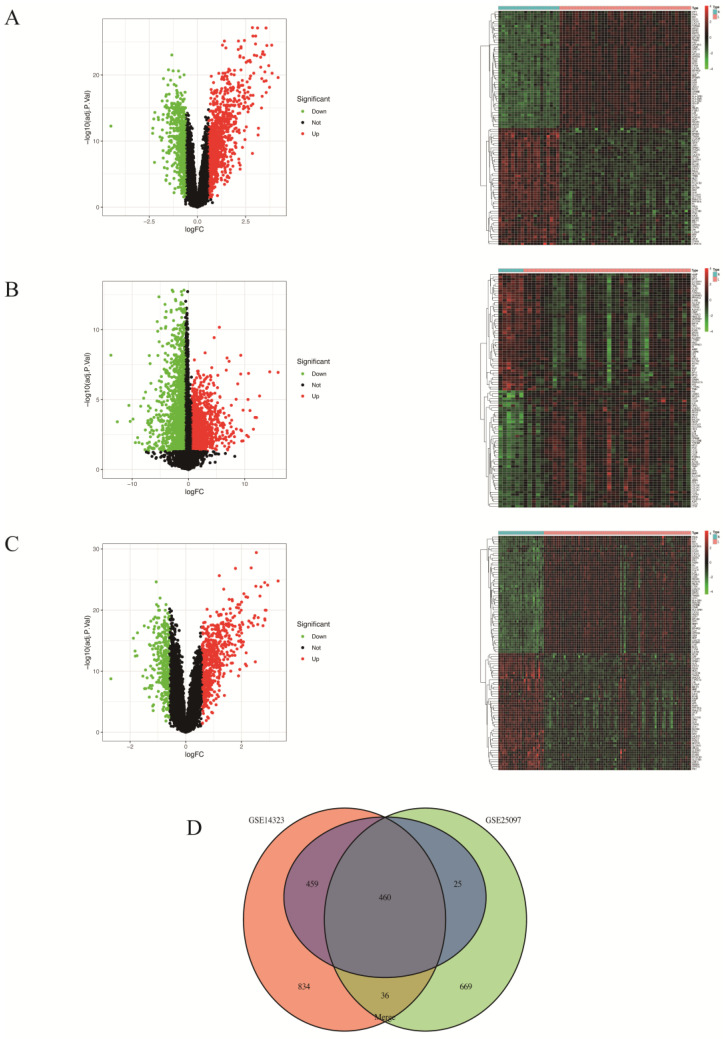
Volcano, heatmap and Venn diagram: (**A**) The volcano map and heatmap of GSE14323. (**B**) The volcano map and heatmap of GSE25097. (**C**) The volcano map and heatmap of the merged dataset. Upregulated genes are marked in light red and downregulated genes are marked in light green. (**D**) The three datasets show an overlap of 460 co-DEGs.

**Figure 2 biomolecules-14-00013-f002:**
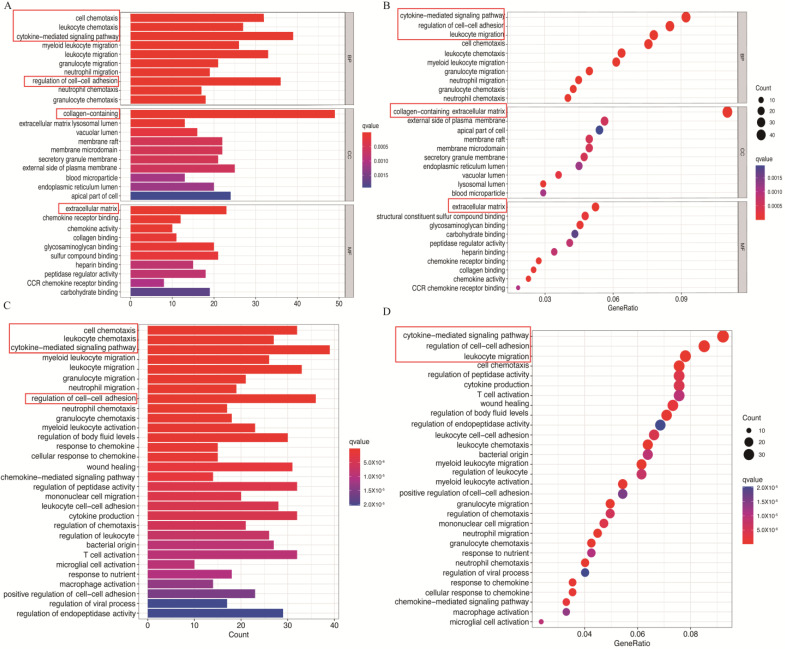
Co-DEGs enrichment analysis results: The enrichment analysis results of the GO (**A**,**B**) and the KEGG (**C**,**D**) analyses. Adjusted *p* value < 0.05 was considered significant. The red boxes contain the key pathways.

**Figure 3 biomolecules-14-00013-f003:**
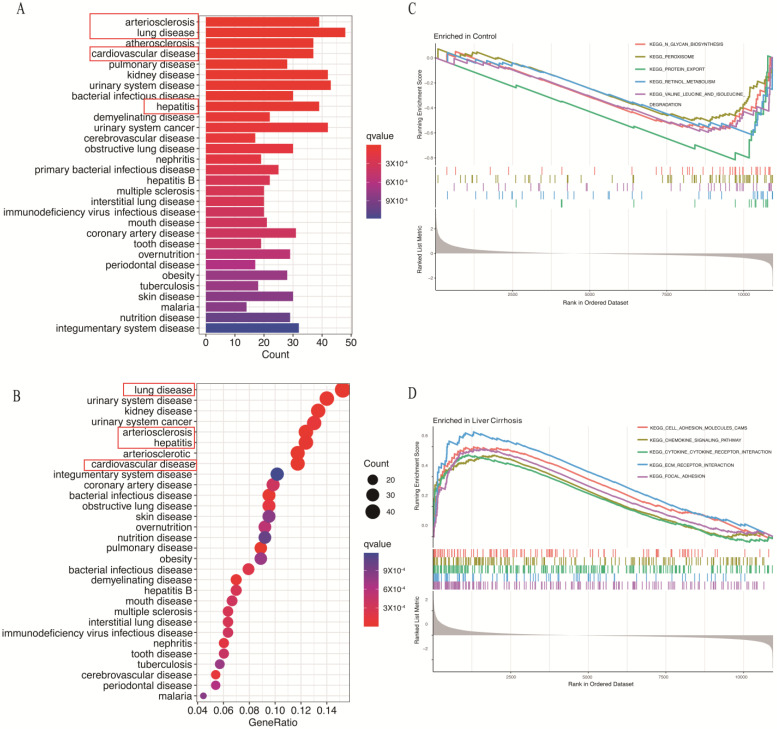
Co-DEGs enrichment analysis results: (**A**,**B**) DO analysis. (**C**,**D**) GSEA analysis. The red boxes contain the main related disease.

**Figure 4 biomolecules-14-00013-f004:**
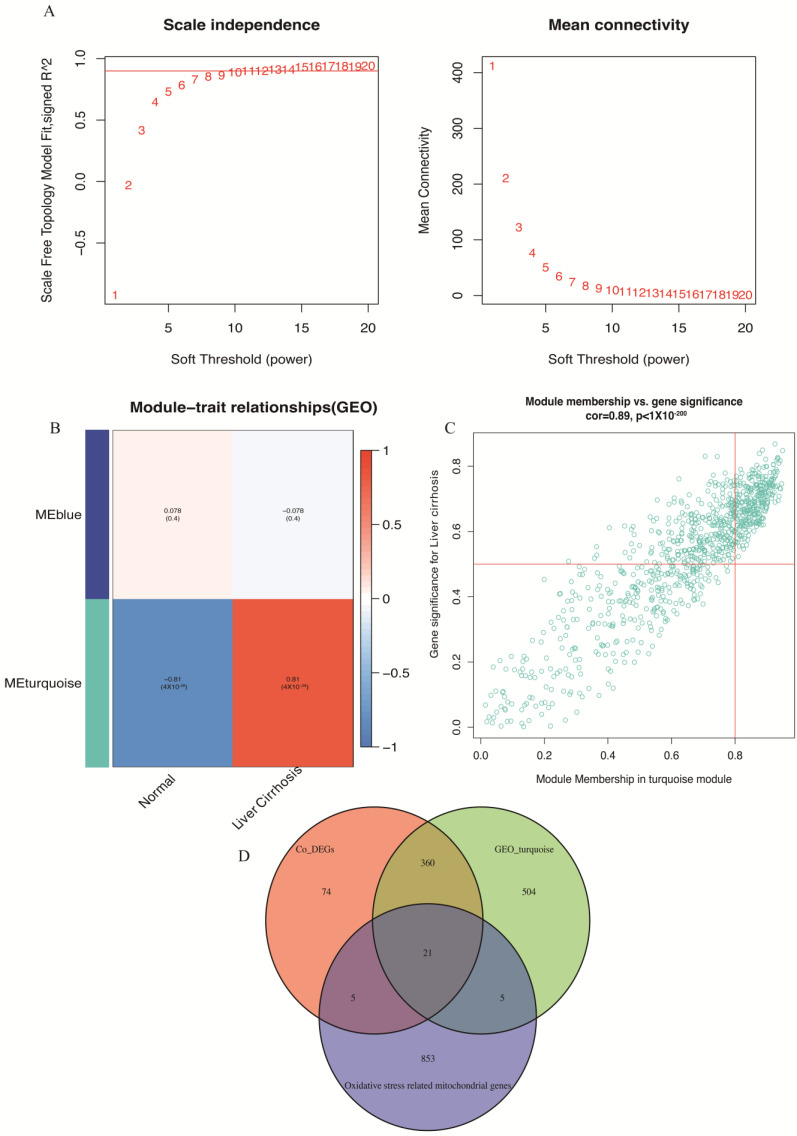
Identifying and screening the related genes: (**A**–**C**) WGCNA analysis. (**D**) The Venn diagram result.

**Figure 5 biomolecules-14-00013-f005:**
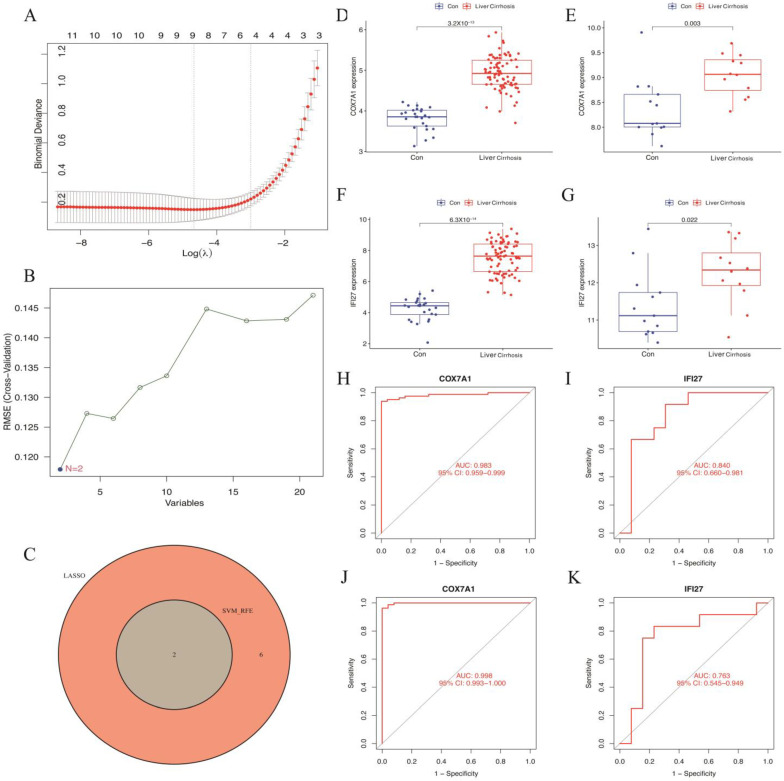
Identifying and verifying the diagnostic identifying genes: (**A**) LASSO logistic regression. (**B**) SVM-RFE algorithm. (**C**) Venn diagram of the intersection of screened diagnostic identifying genes. (**D**,**E**) Gene expression levels in the training set (merged dataset). (**F**,**G**) Gene expression levels in the validation dataset (GSE89377). (**H**,**I**) The ROC analysis of *COX7A1* and *IFI27* in the merged dataset. (**J**,**K**) The ROC analysis of *COX7A1* and *IFI27* in the GSE89377.

**Figure 6 biomolecules-14-00013-f006:**
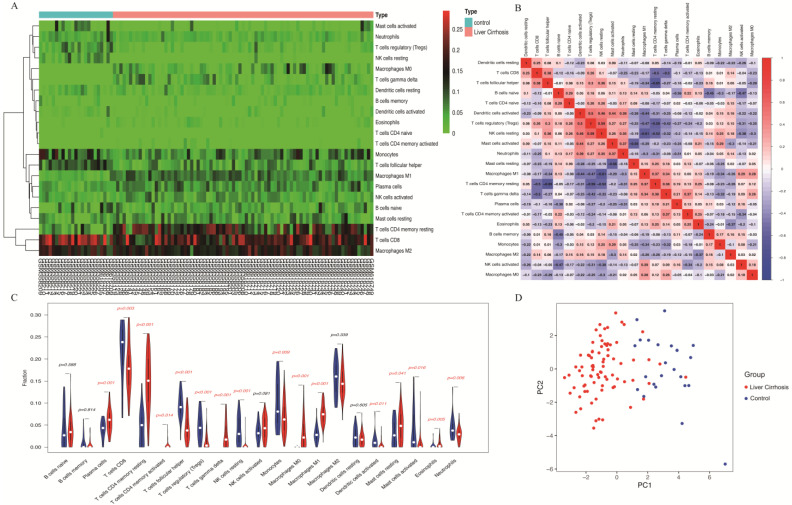
Immune infiltration analysis: (**A**) The heatmap of immune cell infiltration in the normal group (blue) and the liver cirrhosis group (red). (**B**) The correlation heatmap of immune cell concentrations. (**C**) The difference analysis of immune infiltration in the normal group (blue) and the liver cirrhosis group (red). (**D**) PCA of the control group and the liver cirrhosis group.

**Figure 7 biomolecules-14-00013-f007:**
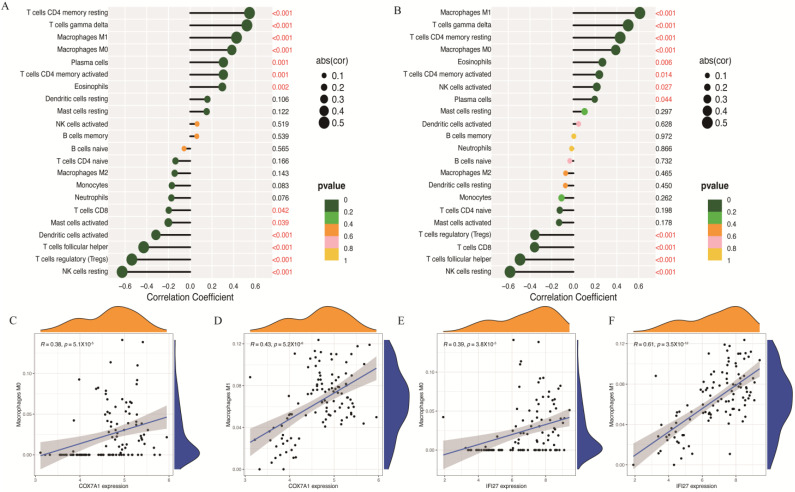
The correlation between two diagnostic identifying genes and different immune cells: (**A**,**B**) The results of the correlation between COX7A1 and *IFI27* and 22 immune cells. (**C**,**D**) The scatterplots of the correlation between COX7A1 and M0 and M1. (**E**,**F**) The scatterplots of the correlation between *IFI27* and M0 and M1. The red color data mean *p* < 0.05.

**Figure 8 biomolecules-14-00013-f008:**
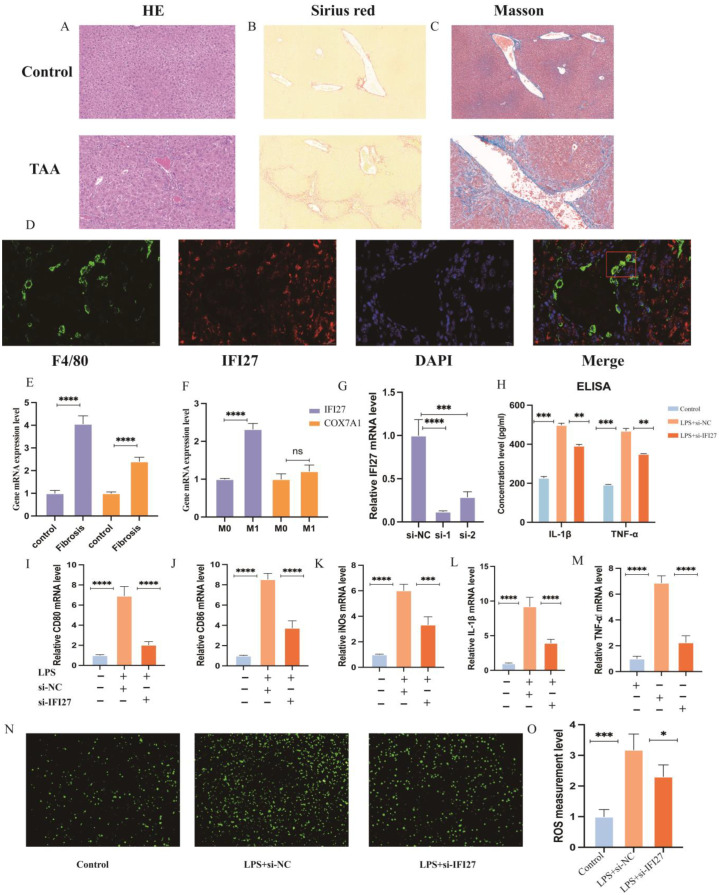
(**A**–**C**) Representative H&E staining, Masson staining and Sirius red staining of mice in the control group and the TAA group, scale: 100 µm. (**D**) Liver tissue immunofluorescence shows co-localization of *IFI27* and liver macrophages. (**E**,**F**) The expression of *COX7A1* and *IFI27* in liver tissue and macrophages. (**G**) After treatment with si-*IFI27*, the knockdown efficiency was detected using RT-PCR. (**H**) ELISA was used to detect the concentration of IL-1β and TNF-α secreted from RAW264.7 in the different groups. (**I**–**M**) mRNA expression of M1 markers (CD80, CD86, iNOs) and pro-inflammation factors (IL-1β and TNF-α) in three groups were assayed using RT-qPCR. Data are presented as means ± SEM. * *p* < 0.05; ** *p* < 0.01; *** *p* < 0.001; **** *p* < 0.0001; ns, not significant. COX7A1, cytochrome c oxidase subunit 7A1; *IFI27*, interferon alpha-inducible protein 27; IL-1β, interleukin-1 beta; TNF-α, tumor necrosis factor alpha. (**N**) ROS generation level of RAW264.7 cells after LPS and si-*IFI27* intervention (scale bar: 200 μm). (**O**) Quantification of ROS-positive area.

**Table 1 biomolecules-14-00013-t001:** The sequence of primers used in this research.

Gene	Forward Sequence (5′-3′)	Reverse Sequence (5′-3′)
GAPDH	GCAGTGCCAGGTGAAAATCG	TACGGCCAAATCCGTTCACA
CD86	AAGGACATGGGCTCGTATGA	GTGACCTTGCTTAGACGTGC
CD80	CAATACGACTCGCAACCACA	CGACTCTTATTACTGCGCCG
iNOS	AATGCCCGTACCAGGCCCAAT	GGTCACCTACCGCACCCGAGAT
IL-1β	TGCCACCTTTTGACAGTGATG	TTCTTGTGACCCTGAGCGAC
TNF-α	TAGCCCACGTCGTAGCAAAC	ACCCTGAGCCATAATCCCCT
*COX7A1*	ATGCCTAACCTAAACATGCCAG	TACTGGGAGGTCATTGTCGG
*IFI27*	TGAGTTCTCCAGAGCCAAGG	GAGCCCACGATGACAGTAGA

## Data Availability

In this study, all data came from publicly available databases and references to available data are included in the methodology section.

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
