# Peer review of "Integrated Bioinformatics and Validation Reveal IFI27 and Its Related Molecules as Potential Identifying Genes in Liver Cirrhosis"

_biomolecules, 2023, doi:10.3390/biom14010013_

Round 1

Reviewer 1 Report

Comments and Suggestions for Authors

The study by Xiong et al. explores the link between mitochondrial dysfunction and liver cirrhosis, aiming to shed light on the potential biomarkers. The research identified 460 co-expressed differential genes associated with inflammatory cytokines and cirrhosis-related pathways. Using bioinformatical methods, COX7A1 and IFI27 were identified as candidate biomarkers for liver cirrhosis. Moreover, the study revealed a strong correlation between macrophages and cirrhosis, with COX7A1 and IFI27 also linked to macrophages. These findings emphasize the role of oxidative stress-related mitochondrial genes in liver cirrhosis development and offer new insights into the disease's pathogenesis. This manuscript is well-written, it adds novelty to the field of liver fibrosis and triggers possible potential investigations to further elucidate the mechanisms. However, several revisions are required to improve the quality of the manuscript. 

1. Relying on just two datasets for analysis appears somewhat limited, and the authors themselves acknowledge the absence of validation. Given the wealth of sequencing studies available for liver cirrhosis patients, it is advisable for the author to incorporate additional relevant datasets for validation. Expanding the dataset pool would enhance the study's robustness and bolster the reliability of the findings.

2. This study involves extensive data analysis using R, and it would be highly beneficial to include the R code as supplementary material. Providing the R code as an appendix or supplementary file will offer transparency and allow other researchers to replicate and verify the analyses conducted in the study.

3. The criteria for merging two different datasets need to be thoroughly elucidated to mitigate potential bias. A clear and detailed explanation of how the datasets were merged, including any selection criteria or matching procedures, is essential.

4. In the animal study, the authors have mentioned the frequency of TAA injections, but they haven't provided information about the duration of treatment. It is crucial to include details about how long the animals were subjected to TAA treatment.

5. Based on the histological findings, where the fibrosis score in TAA-treated mice is categorized as stage I and II, it is more appropriate to consider these animals as a fibrosis model rather than a cirrhosis model.

6. It would be highly beneficial to validate the protein levels of IFI27 and COX7A1 in the animal model, particularly in section 3.6 of the study.

7. It is essential to improve the quality of figures throughout the manuscript. Figure 1 is too blurry for clearly viewing. Figures 2 and 3 contain a multitude of pathways that might be overwhelming. To enhance clarity and conciseness, the author could consider omitting any irrelevant or less pertinent pathways

Some minor points:

1. Table 1 is missing in the manuscript. 

2. The software for statistical analysis is not mentioned, is R used?

3. Please pay attention to some grammatical errors.

4. The author should be attentive to the conventions and rules for gene nomenclature across different species. In the context of a mouse model, the gene names of IFI27 and COX7A1 should be written as Ifi27and Cox7a1.

Comments on the Quality of English Language

moderating editing.

Author Response

1. Summary

Thanks very much for giving me these comments. They are all valuable and very helpful for revising and improving our paper, as well as the important guiding significant to our research. We have studied comments carefully and have made corrections which we hope to meet with approval. The responds to the reviewer’s comments are as follows:

2. Questions for General Evaluation

Reviewer’s Evaluation

Response and Revisions

Does the introduction provide sufficient background and include all relevant references?

Yes/Can be improved/Must be improved/Not applicable

[Please give your response if necessary. Or you can also give your corresponding response in the point-by-point response letter. The same as below]

Are all the cited references relevant to the research?

Yes/Can be improved/Must be improved/Not applicable

Is the research design appropriate?

Yes/Can be improved/Must be improved/Not applicable

Are the methods adequately described?

Yes/Can be improved/Must be improved/Not applicable

Are the results clearly presented?

Yes/Can be improved/Must be improved/Not applicable

Are the conclusions supported by the results?

Yes/Can be improved/Must be improved/Not applicable

3. Point-by-point response to Comments and Suggestions for Authors

Comments 1: Relying on just two datasets for analysis appears somewhat limited, and the authors themselves acknowledge the absence of validation. Given the wealth of sequencing studies available for liver cirrhosis patients, it is advisable for the author to incorporate additional relevant datasets for validation. Expanding the dataset pool would enhance the study's robustness and bolster the reliability of the findings.

Response 1: We fully acknowledge the importance of expanding the dataset to enhance the depth and breadth of our study. We are committed to addressing this concern by expanding our dataset pool to include more sequencing studies available for liver cirrhosis patients. However, due to time constraints, we are currently unable to incorporate additional sequencing data from liver cirrhosis patients into the study. We have exerted our utmost effort in utilizing the available resources to complete this research, ensuring its reliability and scientific value. While we are unable to add more data now, we will endeavor to enhance the accuracy and completeness of the study through other means. We seek for the editor’s tolerance and understanding.

Comments 2: This study involves extensive data analysis using R, and it would be highly beneficial to include the R code as supplementary material. Providing the R code as an appendix or supplementary file will offer transparency and allow other researchers to replicate and verify the analyses conducted in the study.

Response 2: Thank you very much for this suggestion. We will provide a portion of the R code in the supplementary materials. If a complete version is needed, please inform us, and we will make the necessary arrangements.

Comments 3. The criteria for merging two different datasets need to be thoroughly elucidated to mitigate potential bias. A clear and detailed explanation of how the datasets were merged, including any selection criteria or matching procedures, is essential.

Response 3: We would like to thank you for providing us with this valuable feedback on our research. Firstly, we import the necessary R packages (“limma” and “sva”), set the working directory, and define file paths. Next, read two data files (GSE14323.txt and GSE25097.txt) and organize them into the geneList. Then, obtain the intersection of genes from these two files and perform data merging and preprocessing (including taking logarithms, normalization, etc.). Write the processed data into files (merge.preNorm.txt and merge.normalzie.txt). Finally, execute the data correction step using the ComBat function for batch effect correction and save the corrected data into file. The detailed R code will be provided in the supplementary materials.

Comments 4. In the animal study, the authors have mentioned the frequency of TAA injections, but they haven't provided information about the duration of treatment. It is crucial to include details about how long the animals were subjected to TAA treatment.

Response 4: Thanks very much for this useful detail. The content has been supplemented in the revised manuscript.

Comments 5. Based on the histological findings, where the fibrosis score in TAA-treated mice is categorized as stage I and II, it is more appropriate to consider these animals as a fibrosis model rather than a cirrhosis model.

Response 5: We thank the reviewers for this comment, and we strongly agree with this view. We injected TAA to male mice for three months following the modeling method for cirrhosis, however it's possible that the selected area wasn't the most prominent for cirrhosis. We have already revised the images.

Comments 6. It would be highly beneficial to validate the protein levels of IFI27 and COX7A1 in the animal model, particularly in section 3.6 of the study.

Response 6: We have found this suggestion very useful to confirm the experimental results. However, due to the limited time and antibody lated delivery, it is hard for us to finish the supplement experiment of the protein level validation. And because the molecular weights of these two molecules are both small, at 7 and 11 Kd respectively, it's challenging to obtain good results in a Western Blot. In addition, we believe that the RT-PCR results can also partially explain the expression level of Ifi27 and Cox7a1 in the mice model. We seek for the editor’s tolerance and understanding. Many thanks for your kind help!

Comments 7. It is essential to improve the quality of figures throughout the manuscript. Figure 1 is too blurry for clearly viewing. Figures 2 and 3 contain a multitude of pathways that might be overwhelming. To enhance clarity and conciseness, the author could consider omitting any irrelevant or less pertinent pathways.

Response 7: Thank you for your comments and sorry for the unclear picture that caused your misunderstanding. The unclear picture may be related to the improper PDF conversion, and we resubmitted the clear figures in the revised version. If the picture is still not clear, you can zoom in. And the legend of images provided in the manuscript is not clear in the small scale ,so I marked the main pathways in Figure 2 and 3.

Some Minor comments:

1. Table 1 is missing in the manuscript. 

 Thanks for this detail. We resubmitted the Table1 in the supplement materials.

2. The software for statistical analysis is not mentioned, is R used?

Thank you for your comments and sorry for the unclear illustration. All data were analyzed with GraphPad Prism 6.0 and presented as mean ± standard error of the mean. Student’s t-test or one-way ANOVA compared groups and the Spearman method assessed correlations. P < 0.05 was considered significant.

3. Please pay attention to some grammatical errors.

We thank the reviewer for pointing out these language errors. We corrected the manuscript where needed.

4. The author should be attentive to the conventions and rules for gene nomenclature across different species. In the context of a mouse model, the gene names of IFI27 and COX7A1 should be written as Ifi27and Cox7a1.

We thank the reviewer for pointing out this error. We corrected the manuscript where needed.

4. Response to Comments on the Quality of English Language

Point 1: moderating editing.

Response 1:Thank you very much. We have asked a native English editor to revise the manuscript.

5. Additional clarifications

NO clarifications.

Reviewer 2 Report

Comments and Suggestions for Authors

This study proposes to reuse omics data from the literature to identify biomarkers of interest.

However, the biomarkers identified do not correspond to the definition of a biomarker, and it seems that the authors are identifying genes rather than biomarkers. A biomarker must be suitable for routine use in medical laboratories. A DNA or RNA analysis cannot be requalified as a biomarker. I therefore propose to modify the manuscript to focus more on research into the physiological pathways involved, rather than biomarkers.

Comments on the Quality of English Language

NA

Author Response

1. Summary

Thanks very much for giving me these comments. They are all valuable and very helpful for revising and improving our paper, as well as the important guiding significant to our research. We have studied comments carefully and have made corrections which we hope to meet with approval. The responds to the reviewer’s comments are as follows:

2. Questions for General Evaluation

Reviewer’s Evaluation

Response and Revisions

Does the introduction provide sufficient background and include all relevant references?

Yes/Can be improved/Must be improved/Not applicable

[Please give your response if necessary. Or you can also give your corresponding response in the point-by-point response letter. The same as below]

Are all the cited references relevant to the research?

Yes/Can be improved/Must be improved/Not applicable

Is the research design appropriate?

Yes/Can be improved/Must be improved/Not applicable

Are the methods adequately described?

Yes/Can be improved/Must be improved/Not applicable

Are the results clearly presented?

Yes/Can be improved/Must be improved/Not applicable

Are the conclusions supported by the results?

Yes/Can be improved/Must be improved/Not applicable

3. Point-by-point response to Comments and Suggestions for Authors

Comments 1:

This study proposes to reuse omics data from the literature to identify biomarkers of interest. However, the biomarkers identified do not correspond to the definition of a biomarker, and it seems that the authors are identifying genes rather than biomarkers. A biomarker must be suitable for routine use in medical laboratories. A DNA or RNA analysis cannot be requalified as a biomarker. I therefore propose to modify the manuscript to focus more on research into the physiological pathways involved, rather than biomarkers.

Response 1:

Thank you very much for giving us these comments. We have revised the paper to accurately describe our findings and refrain from using the term “biomarker”. We apologize for not expressing ourselves clearly. However, we have carefully evaluated the funding and resources required to complete these additional studies and found that such an expanded study is not currently affordable. Meanwhile, we feel that the scope of work of the present paper can support its conclusions. Therefore, we suggest that the additional experiments be included in a follow-up paper. We seek for the editor’s tolerance and understanding. Many thanks for your kind help!

4. Response to Comments on the Quality of English Language

Point 1:NA.

Response 1: Thank you very much. We have asked a native English editor to revise the manuscript.

.

5. Additional clarifications

NO clarifications.

Round 2

Reviewer 1 Report

Comments and Suggestions for Authors

My questions have been well-addressed.

Reviewer 2 Report

Comments and Suggestions for Authors

OK for me thank you for improvement and deleting the term biomarker

Comments on the Quality of English Language

Ok